**Subject Category:**
Biology (whole organism)

health and disease and epidemiology/ecology

Microsporidia, Lithodidae, crab fishery, pathogens, Magellan region

**Author for correspondence:**
S. M. Rodríguez
e-mail: saramrodriz@gmail.com

# Widespread infection of *Areospora rohanae* in southern king crab (*Lithodes santolla*) populations across south Chilean Patagonia

S. M. Rodríguez[1], J. C. Uribe[4], S. A. Estay[2,7], M. Palacios[3,5,6], R. Pinochet[3,6], S. Oyarzún[5] and N. Valdivia[1,6]

[1]Instituto de Ciencias Marinas y Limnológicas, Facultad de Ciencias, [2]Instituto de Ciencias Ambientales y Evolutivas, Facultad de Ciencias, and [3]Programa de Doctorado en Biología Marina, Facultad de Ciencias, Universidad Austral de Chile, Valdivia, Chile
[4]Instituto de la Patagonia, and [5]Facultad de Ciencias, Universidad de Magallanes, Punta Arenas, Chile
[6]Centro de Investigación Dinámica de Ecosistemas Marinos de Altas Latitudes (IDEAL), Valdivia-Punta Arenas, Chile
[7]Center of Applied Ecology and Sustainability (CAPES), Pontificia Universidad Católica de Chile, Santiago, Chile

 SMR, 0000-0002-1855-4170; JCU, 0000-0002-1295-7796

Cottage cheese disease is caused by microsporidian parasites that infect a wide range of animal populations. Despite its potential to affect economically important activities, the spatial patterns of prevalence of this disease are still not well understood. Here, we analyse the occurrence of the microsporidian *Areospora rohanae* in populations of the king crab *Lithodes santolla* over *ca* 800 km of the southeastern Pacific shore. In winter 2011, conical pots were deployed between 50 and 200 m depth to capture crabs of a wide range of sizes. The infection was widely distributed along the region, with a mean prevalence of 16%, and no significant association between prevalence and geographical location was detected. Males, females and ovigerous females showed similar prevalence values of 16.5 (13–18.9), 15 (9.2–15) and 16.7% (10–19%), respectively. These patterns of prevalence were consistent across crab body sizes, despite the ontogenetic and sex-dependent variations in feeding behaviour and bathymetric migrations previously reported for king crabs. This study provided the first report of the geographical distribution of *A. rohanae* infecting southern king crabs.

# 1. Introduction

Pathogenic parasites can imperil diverse commercial activities, ranging from honeybee culture to fish and crustacean fisheries [1,2]. In recently described pathogenic parasites [3,4], the understanding of the pattern of infection is a mandatory first step to develop powerful predictive models. Considering that host attributes like sex and size may significantly influence the probability of infection of parasites [5,6], the assessment of the pathogenic potential of these species should include the associations (or lack thereof) between prevalence, host sex and host size.

Host sex may influence parasite prevalence in host populations [7–9]. Hormone concentrations, behaviour, immune responses and diet could make female and male individuals behave as distinct types of hosts [9,10]. However, sex-dependent parasitosis can be less evident in systems characterized by parasites with high dispersal potential, because differences between males and females would have no effect on infection probabilities [8,11]. In addition, extreme over-dispersion of parasites across the host population decreases the parasite's ability to control the host population [12], which can lead to indistinct patterns of infection of male and female hosts [9,13]. Disentangling the relationship between sex and prevalence is particularly relevant in those cases where hosts are selectively exploited. For example, several crab fisheries focus on large males, so pathogen-induced mortality would have differential effects on the economic activity if parasitosis is either sex-dependent or independent [3,14].

Across host sex, host body size can associate with parasitosis. In the case of pathogenic parasites, enhanced host mortality leads to a reduction in prevalence in larger, over-infected individuals [15,16]. This, in turn, is evidenced in a unimodal relationship between prevalence and host body size [17]. Yet, there are also examples in which parasitosis prevalence is independent of body size. For instance, probability of infection can show low variability across host ontogeny when the infection occurs primarily in aggregations of juvenile hosts and the parasite has a high transmission efficiency [16].

Cottage cheese disease is caused by microsporidians that infect a wide range of animals in terrestrial and marine ecosystems [1,18]. These parasites develop massive accumulations of spores that destroy and replace the host's muscular tissue—once the musculature is fully replaced, the disease expands to other organs and kills the host [19]. In general, microsporidian parasites can have a high pathogenicity, and when they affect commercially important hosts such as wild populations of crustaceans under exploitation [3,14,19], the spread of the disease can have severe societal consequences. In this line, lithodid crabs host a limited number of microsporidian parasites and sustain important fisheries, with 2000 and 4000 t yr$^{-1}$ captured in the Northern and Southern Hemisphere, respectively [20,21]. In south Chilean fjords, Stentiford *et al.* [1] recently recorded the novel microsporidian *Areospora rohanae*, parasitizing individuals of the king crabs *Lithodes santolla* Molina, 1782. The life history of *A. rohanae* and its relationship with king crab individual attributes are still unknown.

In this study, we evaluate the occurrence of *A. rohanae* in populations of the southern king crab *L. santolla* across 800 km of southern Chilean Patagonian and Magellan shores. We test the prediction that host sex and body size are associated with parasite prevalence across the region. As a null prediction, we predicted a unimodal relationship between prevalence and host body size. Considering that fishing is restricted by law to large (12 cm cephalothorax length) male individuals [22], sex- and size-dependent parasitosis would have important economic consequences in the region. To our best knowledge, this is the first study on the geographical patterns of prevalence of *A. rohanae* microsporidian in king crabs along the south Chilean Patagonia.

# 2. Material and methods

## 2.1. Study sites and sampling procedure

Fifty locations along the southern Chilean Patagonia shore were sampled during winter 2011, spanning 800 km of the coast (50°–56° S; figure 1). The samples were obtained in three oceanographic cruises. In each location, we used conical pots to capture the king crabs. The pots were moored for 48 h between 50 and 200 m deep to capture a wide range of crab sizes [23,24]. The individuals were transferred alive to the Centro de Investigación de Recursos Marinos de Ambientes Subantárticos (CERESUB) at the Universidad de Magallanes. In the laboratory, the crabs were subjected to a thermal shock at −80°C and kept at −40°C. Later, the carapace length of each crab was measured. Also, the individuals were

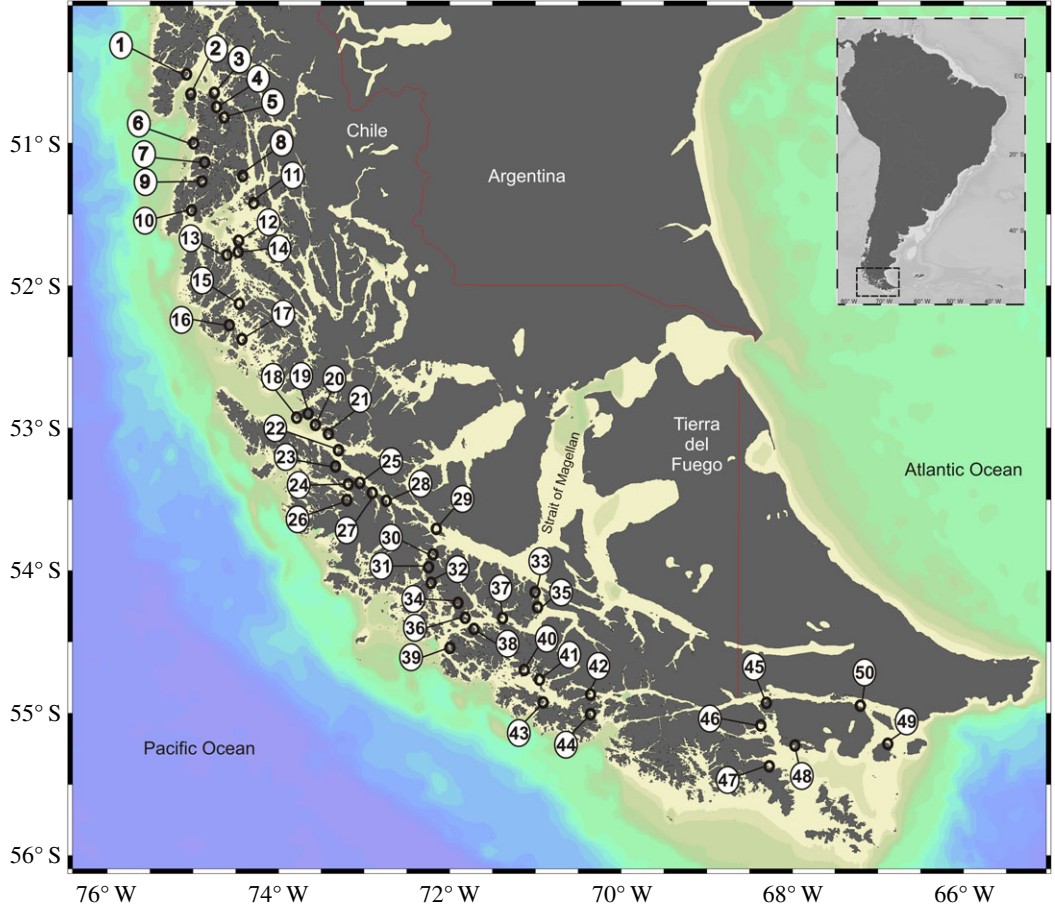

**Figure 1.** Study sites of *Lithodes santolla* in Chilean south Patagonia (Magellan region). 1. Isla Saboya, 2. Isla Toro, 3. Canal Rayo, 4. Canal Farrel 2, 5. Canal Farrel 1, 6. Canal Ignacio, 7. Canal Guadalupe, 8. Isla Sofía, 9. Canal San Blas, 10. Seno de los Torrentes, 11. Grupo Solari, 12. Isla Wilson, 13. Isla Torres, 14. Canal Uribe, 15. Canal Ballena, 16. Canal Bertrand, 17. Isla Summer, 18. Paso Roda, 19. Bahía Monsón, 20. Isla Providencia, 21. Isla Richarson, 22. Isla Santa Ana, 23. Estero indio, 24. Canal Abra, 25. Isla Childs, 26. Isla Larga, 27. Seno de las Nieves, 28. Estero Nevado, 29. Isla Charles, 30. Isla Alcayata, 31. Isla Browell, 32. Bahía Brown, 33. Puerto Hope, 34. Isla Julio, 35. Isla Laberinto, 36. Seno Dounze, 37. Isla King, 38. Seno Brujo, 39. Canal Ocasión, 40. Seno los Ladrones, 41. Canal Ballenero, 42. Grupo Timbales, 43. Isla Stewart, 44. Isla Luisa, 45. Puerto Navarino, 46. Isla Mascart, 47. Bahía Navidad, 48. Isla Bertrand, 49. Isla Lennox, 50. Puerto Eugenia.

separated into male, ovigerous females (abdomen cavity with eggs) and non-ovigerous females (abdomen cavity without eggs; [25]).

For each crab, we registered the presence or absence of *A. rohanae* infection. The infection was evidenced by whitish nodules arising from the sub-cuticular tissues in multiple parts of the body and thoracic appendages (figure 2; [1]). Stentiford *et al*. [1] described and corroborated the microsporidiosis by means of PCR, ultrastructure descriptions and histological analyses (figure 2). For each sex group, the individuals were separated in twelve 1 cm size classes. The entire size range was of 2–19 cm carapace length. For each group, the prevalence was estimated as the proportion of infected hosts [26].

## 2.2. Statistical analyses

Region-level prevalence and confidence intervals were estimated by means of 1000 bootstrapped samples of the original sample. The same method was used to estimate the sex-specific proportion of infected hosts.

We used a generalized linear mixed modelling (GLMM) approach to analyse the infection as a function of crab length, sex and location. Length and sex were included in the model as fixed factors; location (i.e. the interaction of latitude and longitude) was included as a random factor, influencing the effects of both fixed factors on the probability of infection (i.e. a random-slope model). The model

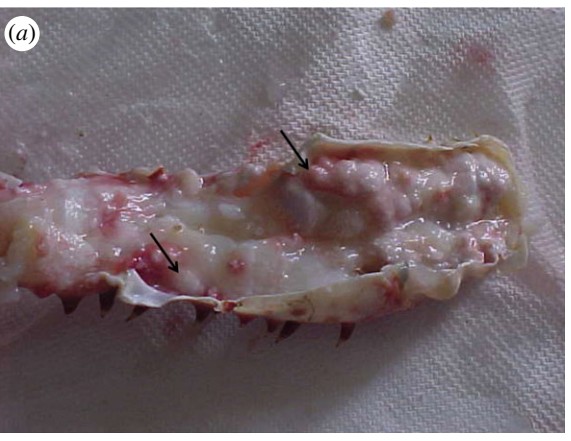

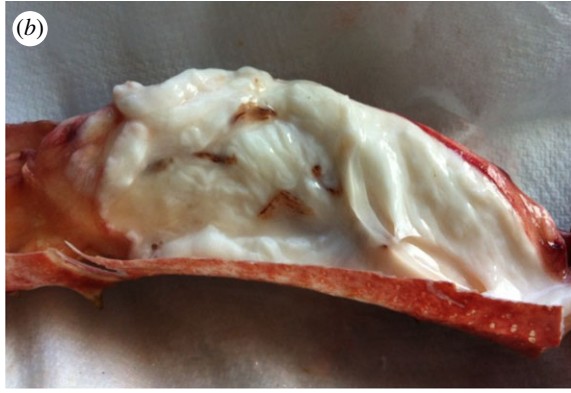

**Figure 2.** Tissue of *Lithodes santolla* infected by *Areospora rohanae*. (*a*) Granules of *A. rohanae* (arrows) in appendage. (*b*) Signs of the 'Cottage cheese disease' present throughout the appendage tissue. Photographs by S. Oyarzún.

also included sample size of each locality in order to control for potential artefacts caused by varying sampling effort across the region. Since our null prediction was a unimodal curve, we used a second-order orthogonal polynomial contrast for the factor 'size'. We used maximum likelihood for parameter estimations. A residual-versus-fitted plot was used as a diagnostic of the global (full) model [27]. Modification of likelihood ratio-based pseudo-$R^2$ was calculated to estimate the variability accounted for by the global model. The null model excluded the random factors, so the estimated pseudo-$R^2$ represented the variability explained by the entire model. In addition, we used posterior predictive simulations of prevalence (1000) to determine if the model actually represented the data [28]—this was corroborated with a *p*-value of 0.48. Autocorrelation of residuals of the global model was revised in an autocorrelation function plot. All statistical analyses were conducted with the packages boot, ggplot2, lme4 and MuMIn in R v. 3.3.0 [29–31].

## 3. Results

A total of 3000 southern king crabs were examined. All stations evidenced the presence of *A. rohanae* across the region, and station-level prevalence ranged between 10 and 30% (figure 3). Region-level parasite prevalence was 16% (95% CI = 15–18.7%; table 1). Sex-dependent values of prevalence were estimated as 16.5% (range: 13–18.9%), 15% (9.2–15%) and 16.7% (10–19%) for males, females and ovigerous females, respectively.

Sex-specific prevalence showed varying patterns across body size. Males exhibited a positive relationship between prevalence and body size (red symbols in figure 4). By contrast, female and ovigerous females showed maximum values of prevalence at intermediate sizes (green and blue symbols in figure 4, respectively)—the prevalence of ovigerous females was skewed toward smaller crabs (figure 4). Despite these patterns, however, the GLMM showed a weak relationship of the analysed factors with prevalence, evidenced by a pseudo-$R^2$ of 0.018. This result suggests body size and sex, in addition to the geographical position of the sampling location had a very low predictive power of parasite infection.

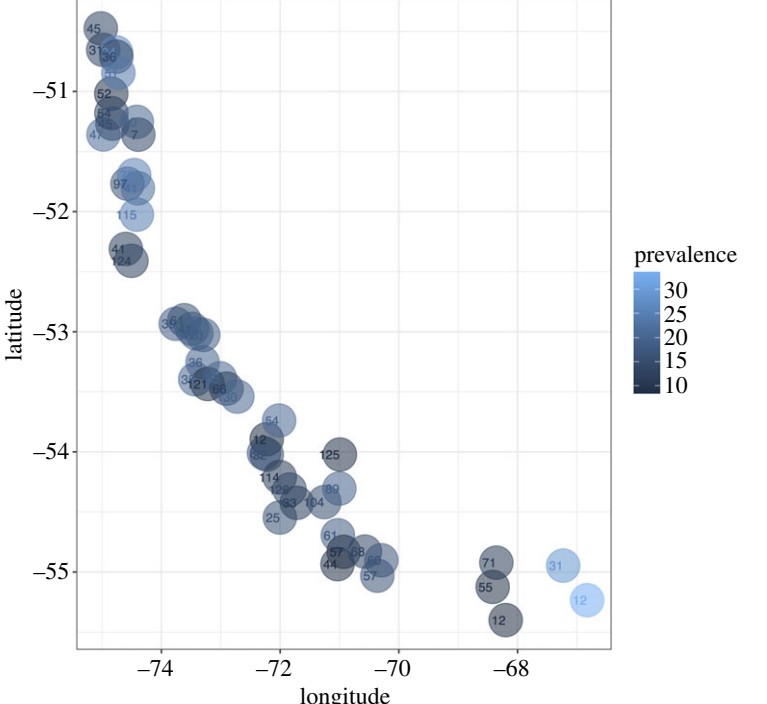

**Figure 3.** Prevalence of *Areospora rohanae* in king crab *Lithodes santolla* along south Chilean Patagonia. Colour scale indicates parasite prevalence in each sampled location. The number of crabs in each sample is provided.

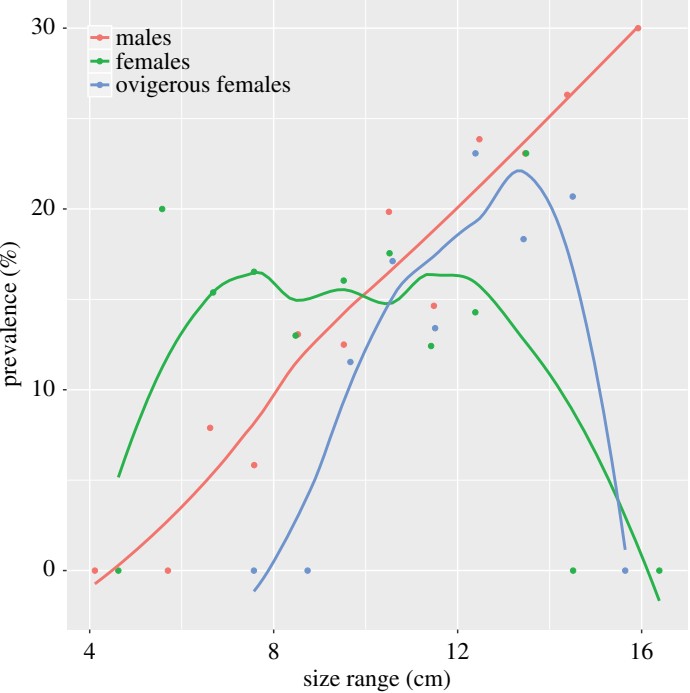

**Figure 4.** Prevalence (% infected) of infection by *A. rohanae* as a function of host size (cm) and sex. The red, green and blue lines indicate male, female and ovigerous female hosts, respectively. Lines were obtained by means of local polynomial regression smoothing (loess).

## 4. Discussion

The results of this study showed that *A. rohanae* infection in southern king crabs was comparatively high across south Chilean Patagonia. Moreover, the prevalence was similar among sampling stations, between

**Table 1.** Prevalence (% infected) of infection by *A. rohanae* for sex in each sampled location. The sign '–' indicates absence of individuals.

| site | prevalence (%) | | | |
| --- | --- | --- | --- | --- |
| | total | male | female | ovigerous female |
| Canal Uribe | 41 | 40 | 0 | 24.1 |
| Seno de los Torrentes | 47 | 30.8 | 10 | 0 |
| Canal Rayo | 31 | 20 | 0 | 0 |
| Grupo Solari | 7 | 20 | 0 | – |
| Isla Sofía | 55 | 14 | 17.6 | 0 |
| Isla Toro | 34 | 38.5 | 8.3 | 33.3 |
| Canal Ignacio | 52 | 10 | 4.8 | 18.2 |
| Canal Guadalupe | 54 | 14 | 8 | 50 |
| Canal Farrel 1 | 51 | 33.3 | 0 | 25 |
| Canal Farrel 2 | 36 | 12.5 | 15.8 | 0 |
| Canal San Blas | 45 | 1.1 | 16.7 | 33.3 |
| Isla Torres | 34 | 38.5 | 8.3 | 33.3 |
| Canal Ballena | 45 | 9.5 | 23.5 | 14.3 |
| Canal Bertrand | 70 | 20 | 32.4 | 50 |
| Isla Wilson | 72 | 29.2 | 23.3 | 20 |
| Isla Summer | 124 | 16.7 | 9.8 | 0 |
| Paso Roda | 39 | 16.7 | 0 | 25 |
| Isla Charles | 54 | 8 | 31 | – |
| Bahía Brown | 32 | 15.4 | 14.3 | 16.7 |
| Estero Nevado | 130 | 8.1 | 24.3 | 25 |
| Isla Providencia | 41 | 33.3 | 0 | 16.7 |
| Seno de la Nieves | 66 | 12.1 | – | – |
| Isla Richardson | 42 | 19 | 20 | 18.8 |
| Bahía Monson | 61 | 26.5 | 6.2 | 0 |
| Isla Childs | 85 | 29.7 | 12.5 | 12.5 |
| Canal Abra | 35 | 14.3 | 26.7 | 16.7 |
| Isla Santa Ana | 120 | 12.9 | 30.4 | 33.3 |
| Estero Indio | 36 | – | 26.3 | 11.8 |
| Isla Larga | 121 | 14.5 | 7.5 | 0 |
| Isla Julio | 114 | 8.1 | 21.2 | 5.3 |
| Isla Browell | 80 | 16.7 | 18.8 | 16.7 |
| Isla Alcayaga | 12 | 10 | 0 | 0 |
| Bahía Latorre | 44 | 9.5 | 9.1 | 0 |
| Bahía Navidad | 12 | 0 | 14.3 | 0 |
| Canal Ocasión | 25 | 16.7 | 15.4 | – |
| Canal Pomar | 68 | 11.5 | 0 | 17.6 |
| Grupo del Medio | 57 | 6.2 | 12 | – |
| Grupo Timbales | 60 | 18.5 | 6.7 | 22.2 |
| Isla Bertrand | 41 | 8 | 18.8 | – |
| Isla King | 104 | 21.2 | 15.9 | 7.4 |

(*Continued.*)

**Table 1.** (*Continued.*)

| site | prevalence (%) | | | |
|------|-------|------|--------|-----------------|
|      | total | male | female | ovigerous female |
| Isla Laberinto | 89 | 24.3 | 15.4 | – |
| Isla Lennox | 12 | 12.5 | 75 | – |
| Isla Luisa | 57 | 46.2 | 0 | 8.8 |
| Isla Mascart | 55 | 11.5 | 7.1 | 0 |
| Puerto Eugenia | 31 | 33.3 | 14.3 | – |
| Puerto Hope | 125 | 7.8 | 8.6 | 33.3 |
| Puerto Navarino | 71 | 10 | 9.4 | 15.8 |
| Seno Brujo | 33 | 50 | 0 | 7.4 |
| Seno Dounze | 126 | 19.2 | 11.9 | 9.4 |
| Seno los Ladrones | 61 | 13.6 | 12.5 | 33.3 |

host sexes and along body size. Accordingly, our analysis of statistical modelling suggests that microsporidiosis was independent of geographical location, sex and size of hosts. In this section, we discuss the processes that result in a high and widespread infection of *A. rohanae*, in addition to the potential socio-economic consequences of this disease in Chilean southern Patagonia.

## 4.1. High *A. rohanae* prevalence in southern king crabs

*Areospora rohanae* was widely distributed across the study region with a prevalence of *ca* 16%. This value was higher than those observed for microsporidian parasites infecting wild populations of other crabs elsewhere. In the Sea of Okhotsk (Russia), for instance, *Thelohania* sp. and *Ameson* sp. show prevalence values of 3.2% and 0.2% in their respective hosts [14]. Moreover, the microsporidian *Nadelspora canceri* occurs in 14% of the *Metacarcinus magister* population along the United States Pacific coast [32]. Then, why was the prevalence of *A. rohanae* in the southern king crab comparatively high? The scarcity of published information of how the disease is transmitted makes it difficult to construct testable hypotheses to answer this question. Research conducted on other microsporidians, such as *Thelohania contejeani* and *T. montirivulorum*, suggests multiple ways of infection [33,34] that can contribute to the disease's high prevalence. For instance, the spores can be trophically transmitted to new hosts [19,35]: in the case of crabs, ingestion of spores could occur through scavenging (crayfish: [34]) and cannibalism (crabs: [3,32]; king crabs: [36,37]). Moreover, the microsporidian spores can be dispersed through seawater currents and a single host can release millions of spores [34,35]. Thus, multiple ways of transmission could underpin a high dispersal potential of *A. rohanae*.

Literature indicates that cottage cheese disease is produced by diverse microsporidian species [1,18]. Along with this, other microsporidian species that do not produce this disease have been reported in the soft tissues or sub-cuticular epidermis in the same crab host [1,14,38]. Also, some hosts harbour more than one microsporidian parasite with similar clinical signs of infection. This may suggest that *A. rohanae* symptoms may be confounded with those of other parasites in our study. Also, our use of gross visual assessment may have underestimated the prevalence of *A. rohanae*, because the macroscopic signs used here probably represent late-stage infections [38]. Despite these potential drawbacks of our study, we were able to report a comparatively high prevalence of this parasite, which should be considered in a more detailed research agenda of this recently described parasite in southern Chilean Patagonia. In addition, future research is needed to shed light on potential patterns of co-infection between *A. rohanae* and other microsporidian parasites.

At what size do king crabs get infected? The life history of southern king crabs may provide data on the process of infection. For instance, high individual density triggers agonistic interactions and cannibalism in juvenile *L. santolla*, which can increase the infection probability of small-sized individuals [39,40]. Thus, large aggregations during early stages of king crabs would be generating a 'window of opportunity' for the transmission of *A. rohanae*. Assuming a high pathogenicity of

microsporidians (e.g. in crayfish: [41]), then it would be hypothesized that an early-stage infection of *A. rohanae* implies a mechanism of control of host population growth.

## 4.2. Potential economic impact of microsporidiosis in king crab

What would be the consequences of this infection for king crab fisheries? In the Northern Hemisphere, a sharp decline in crab abundance has been observed in the last years due to microsporidian infections [4,14], notably affecting the economic activity [18,19]. In the Magellan region, *L. santolla* supports the most important fishery and their exports have increased over the years, reaching US$52 million in 2016 [42]. Our results showed that 16% of the population was infected by *A. rohanae*, irrespective of sex and body size. This indicates that a relatively large proportion of fished crab would be infected and thus could not be sold. However, the infection can eventually develop cottage cheese disease, enhancing the mortality rate in the population. Since we sampled only live king crabs, we ignore at this moment the mortality that could be caused by *A. rohanae*. Therefore, further quantitative research is mandatory to assess the potential socio-economic consequences of the *A. rohanae* infection of southern king crab fisheries.

## 5. Conclusion

In summary, our results indicated that *A. rohanae* is widely distributed along south Chilean Patagonia. Prevalence was randomly distributed across an 800 km section of the coast, host sex and ontogeny. Factors such as cannibalism during early life stages would enhance the transmission among juvenile individuals, according to high densities of individuals that can be found in shallow waters. Mortality after early-stage transmission would contribute to the parasite's ability to control the host population growth. Finally, the infection by *A. rohanae* might have important socio-economic consequences for local artisanal fisheries in the southeastern Pacific.

Data accessibility. King crabs data are available within the Dryad Digital Repository: https://doi.org/10.5061/dryad.7hs993m [43].

Authors' contributions. J.C.U. and S.O. conceived and designed the study. J.C.U. managed and coordinated the staff and fieldworks. M.P. and R.P. collected and processed the samples. S.M.R., N.V. and S.A.E. analysed and interpreted the data. S.M.R. and N.V. wrote the paper. All authors read and approved the final manuscript.

Competing interests. The authors declare that they have no competing interests.

Funding. This work was funded by Fondo de Innovación a la Competitividad Regional (FIC-R) (grant no. BIP 30111074-0) provided to J.C.U., S.O. and M.P. N.V., M.P. and R.P. were supported by FONDAP (grant no. 15150003) Centro de Investigación Dinámica de Ecosistemas Marinos de Altas Latitudes (IDEAL). N.V. also acknowledges the support provided by FONDECYT (grant nos. 1190529 and 1181300). S.A.E acknowledges the support of Fondecyt (grant no. 1160370) and CAPES Conicyt PIA/Basal (grant no. FB0002). Postdoctoral Fondecyt (grant no. 3190348) funded S.M.R.

Acknowledgements. The authors are grateful to Universidad de Magallanes, Universidad Austral de Chile and Centro IDEAL.

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
