## [Reviewer comments · Royal Society Open Science]

Review History

RSOS-190682.R0 (Original submission)

Review form: Reviewer 1

Is the manuscript scientifically sound in its present form?

No

Are the interpretations and conclusions justified by the results?

No

Is the language acceptable?

Yes

Is it clear how to access all supporting data?

Not Applicable

Do you have any ethical concerns with this paper?

No

Have you any concerns about statistical analyses in this paper?

I do not feel qualified to assess the statistics

Recommendation?

Reject

Comments to the Author(s)

Major:

This is a simple study that would be better presented as a shorter paper/note in a specialist journal. 57 cited references are overkill for a study that only assessed prevalence over a single sample of crabs from one time point (winter 2011). So little is known of this parasite (described in 2014), particularly its mechanisms of transmission, dispersal, seasonality etc., so most text hypothesizing mechanisms driving prevalence/distribution are supposition at best. I suggest the authors restructure the paper as a short report-the prevalence/distribution data is valuable, but the context needs to be changed.

It is not clear to this reviewer that the pathognomonic sign for infection of *A. rohanae* has been well established and tested, yet the gross presentation has been used as a diagnostic tool in the current study (lines 113-118). I have read Stentiford et al. (2014) study and nowhere in that paper are the number of crabs that were used to describe the parasite presented. PCR data was only generated from two independent samples. Stentiford et al do not indicate that no other infections can be associated to these symptoms in king crabs (incorrectly stated by the authors in lines 117-118). Further, cottage cheese disease (pathogen not described to species level) has previously been reported from king crabs in the northern hemisphere. Clearly further work needs to be carried out to establish, beyond doubt, that the presented symptoms are unique to *A. rohanae* infection.

Minor:

Line 35-36, expand on the stated significant association between prevalence and geographical location as there is no mention of this in the results section, page 8??

Line 41-42, delete as speculation.

Lines 67-68, FYI conversely, many crustaceans show a higher prevalence of pathogens in earlier life stages.

Line 71, rewrite to read "In the case"

Line 81-82, better describe "depolymerization of the contractile apparatus" as I don't understand what the authors mean. How do spores do this?

Line 86, do Lithodid crabs harbor diverse microsporidian parasites? I have only seen very few microsporidian species mentioned in the associated king crab literature? Can the authors elaborate?

Line 105, what is a conical plot? Do the authors not mean "pots"? Provide mesh width of traps/pots.

Line 172-173, there is NO data to suggest that *A. rohanae* transmission is by horizontal mechanisms. Stentiford et al. do not mention this anywhere in their paper. Alternate hosts may well be involved but this has not been studied.

Line 180, a single infected host could release millions of infective spores, not hundreds....

Line 189-190, no where in this study was the presence of *A. rohanae* established other than by gross presentation. It is completely unethical/unscientific to state that "*L. santolla* was infected only with *A. rohanae*" especially as the material the authors worked with was frozen at -80 deg (destroying tissues).

Review form: Reviewer 2 (Vladimir G. Dvoretzky)

Is the manuscript scientifically sound in its present form?

Yes

Are the interpretations and conclusions justified by the results?

Yes

Is the language acceptable?

Yes

Is it clear how to access all supporting data?

Yes

Do you have any ethical concerns with this paper?

No

Have you any concerns about statistical analyses in this paper?

No

Recommendation?

Accept with minor revision (please list in comments)

Comments to the Author(s)

The authors of the manuscript "Widespread infection of *Areospora rohanae* in southern king crab (*Lithodes santolla*) populations across south Chilean Patagonia" have written a fairly clear and concise paper on a relevant topic. However, it needs some clarification. Here, I have recommended a number of minor revisions that should be fixed in a revised version.

L. 33, L. 105. Change "conical plots" to "conical pots".

L. 34, L. 106. Change "an ample range" to "a wide range"

L. 59. Change "lowers" to "decreases"

L. 75 Change "little" to "low"

L. 87 Change "2000 and 4000 tons year⁻¹" to "2,000 and 4,000 t per year"

L. 106 Be consistent: change "traps" to "pots" or "pots" to "traps" throughout the text.

L. 106 Change "50 and 200 meters" to "50 and 200 m"

L. 112 Delete ";

L. 119 Change "ranges" to "classes"

L. 160 Change "societal" to "socio-economic"

L. 164 Delete "and"

L. 185 Change "and anomuran crab (*Eupagurus bernhardus*)" to "and the hermit crab *Eupagurus bernhardus*"

- L. 189 Change "pathogenesis they" to "their pathogenesis"
- L. 196 Change "may shed some light" to "may provide data"
- L. 211-212 Change "region L. santolla is the most important fishery" to "region, L. santolla support the most important fishery".
- L. 217 Change "living king crabs" to "live king crabs"
- L. 219 Change "socioeconomical" to "socio-economic"
- L. 345 Change "Fish B-NOAA" to "Fish. Bull."
- L. 413 Delete "in."
- Page 24. In my opinion, for better presentation, the authors should delete the number of individuals for each size range and sex from their Figure 1. Optionally, this information can be provided in a separate table (in Materials and Methods).

Review form: Reviewer 3

Is the manuscript scientifically sound in its present form?

Yes

Are the interpretations and conclusions justified by the results?

Yes

Is the language acceptable?

Yes

Is it clear how to access all supporting data?

No

Do you have any ethical concerns with this paper?

No

Have you any concerns about statistical analyses in this paper?

No

Recommendation?

Accept with minor revision (please list in comments)

Comments to the Author(s)

I think that it would have been necessary to examine sub samples of L. santolla by histology, including those do not display clinical signs of infection. Clinical signs as whitish nodules arising from the sub-cuticular tissues are macroscopic signs of probably late-stage/patent infections (e.g. Field et al. 1992, Shields & Behringer 2004, Stentiford et al. 2010, 2014), and use of gross visual assessment methods may significantly underestimate the actual prevalence of A. rohanae. May be to save this point (since the sampling was already done), you could little explain in discussion section.

Prevalence associated with geographic locations was examined. Did you study it with respect to depth of samples?

No association between prevalence and 36 geographic location was detected, may be it could be explained by the different sampling sizes at each location (e.g. in Grupo Solari only 7 king crabs were collected whilst in Isla Toro 34 king crabs were collected). It might be discussed in the corresponding section. A table of these results presenting sampling size, sex proportion and their prevalence (besides the map) is suggested to be provided.

A co-habitation experiment with infected crabs and uninfected ones to explore potential for crab-crab transmission could be addressed.

Page7- Line111: Replace "oocytes" by "eggs"

Line 112: Idem

Page 10- Line 165: Replace "larger" by "higher"

What is the commercial size of *L. santolla*? It is suggested to provide this information.

Decision letter (RSOS-190682.R0)

31-Jul-2019

Dear Dr Rodríguez,

The editors assigned to your paper ("Widespread infection of *Areospora rohanae* in southern king crab (*Lithodes santolla*) populations across south Chilean Patagonia") have now received comments from reviewers. We would like you to revise your paper in accordance with the referee and Associate Editor suggestions which can be found below (not including confidential reports to the Editor). Please note this decision does not guarantee eventual acceptance.

Please submit a copy of your revised paper before 23-Aug-2019. Please note that the revision deadline will expire at 00.00am on this date. If we do not hear from you within this time then it will be assumed that the paper has been withdrawn. In exceptional circumstances, extensions may be possible if agreed with the Editorial Office in advance. We do not allow multiple rounds of revision so we urge you to make every effort to fully address all of the comments at this stage. If deemed necessary by the Editors, your manuscript will be sent back to one or more of the original reviewers for assessment. If the original reviewers are not available, we may invite new reviewers.

- Data accessibility

If you wish to submit your supporting data or code to Dryad (<http://datadryad.org/>), or modify your current submission to dryad, please use the following link:
<http://datadryad.org/submit?journalID=RSOS&manu=RSOS-190682>

- Competing interests

- Authors' contributions

- Acknowledgements

- Funding statement

on behalf of Kevin Padian (Subject Editor)
 openscience@royalsociety.org

Subject Editor Comments to Authors:

Given the variety of comments it seems likely that the authors can address the concerns in a reasonable timeframe, but if more is needed, please let us know. Thanks for submitting.

Reviewers' Comments to Author:

Reviewer: 1

Major:

This is a simple study that would be better presented as a shorter paper/note in a specialist journal. 57 cited references are overkill for a study that only assessed prevalence over a single sample of crabs from one time point (winter 2011). So little is known of this parasite (described in 2014), particularly its mechanisms of transmission, dispersal, seasonality etc., so most text hypothesizing mechanisms driving prevalence/distribution are supposition at best. I suggest the authors restructure the paper as a short report-the prevalence/distribution data is valuable, but the context needs to be changed.

It is not clear to this reviewer that the pathognomonic sign for infection of *A. rohanae* has been well established and tested, yet the gross presentation has been used as a diagnostic tool in the current study (lines 113-118). I have read Stentiford et al. (2014) study and nowhere in that paper are the number of crabs that were used to describe the parasite presented. PCR data was only generated from two independent samples. Stentiford et al do not indicate that no other infections can be associated to these symptoms in king crabs (incorrectly stated by the authors in lines 117-118). Further, cottage cheese disease (pathogen not described to species level) has previously been reported from king crabs in the northern hemisphere. Clearly further work needs to be carried out to establish, beyond doubt, that the presented symptoms are unique to *A. rohanae* infection.

Minor:

Line 35-36, expand on the stated significant association between prevalence and geographical location as there is no mention of this in the results section, page 8??

Line 41-42, delete as speculation.

Lines 67-68, FYI conversely, many crustaceans show a higher prevalence of pathogens in earlier life stages.

Line 71, rewrite to read "In the case"

Line 81-82, better describe "depolymerization of the contractile apparatus" as I don't understand what the authors mean. How do spores do this?

Line 86, do Lithodid crabs harbor diverse microsporidian parasites? I have only seen very few microsporidian species mentioned in the associated king crab literature? Can the authors elaborate?

Line 105, what is a conical plot? Do the authors not mean "pots"? Provide mesh width of traps/pots.

Line 172-173, there is NO data to suggest that *A. rohanae* transmission is by horizontal mechanisms. Stentiford et al. do not mention this anywhere in their paper. Alternate hosts may well be involved but this has not been studied.

Line 180, a single infected host could release millions of infective spores, not hundreds....

Line 189-190, no where in this study was the presence of *A. rohanae* established other than by gross presentation. It is completely unethical/unscientific to state that "*L. santolla* was infected only with *A. rohanae*" especially as the material the authors worked with was frozen at -80 deg (destroying tissues).

Reviewer: 2

Comments to the Author(s)

The authors of the manuscript "Widespread infection of *Areospora rohanae* in southern king crab (*Lithodes santolla*) populations across south Chilean Patagonia" have written a fairly clear and concise paper on a relevant topic. However, it needs some clarification. Here, I have recommended a number of minor revisions that should be fixed in a revised version.

L. 33, L. 105. Change "conical plots" to "conical pots".

L. 34, L. 106. Change "an ample range" to "a wide range"

L. 59. Change "lowers" to "decreases"

L. 75 Change "little" to "low"

L. 87 Change "2000 and 4000 tons year⁻¹" to "2,000 and 4,000 t per year"

L. 106 Be consistent: change "traps" to "pots" or "pots" to "traps" throughout the text.

L. 106 Change "50 and 200 meters" to "50 and 200 m"

L. 112 Delete ";

L. 119 Change "ranges" to "classes"

L. 160 Change "societal" to "socio-economic"

L. 164 Delete "and"

L. 185 Change "and anomuran crab (*Eupagurus bernhardus*)" to "and the hermit crab *Eupagurus bernhardus*"

L. 189 Change "pathogenesis they" to "their pathogenesis"

L. 196 Change "may shed some light" to "may provide data"

L. 211-212 Change "region *L. santolla* is the most important fishery" to "region, *L. santolla* support the most important fishery".

L. 217 Change "living king crabs" to "live king crabs"

L. 219 Change "socioeconomical" to "socio-economic"

L. 345 Change "Fish B-NOAA" to "Fish. Bull."

L. 413 Delete "in."

Page 24. In my opinion, for better presentation, the authors should delete the number of individuals for each size range and sex from their Figure 1. Optionally, this information can be provided in a separate table (in Materials and Methods).

Reviewer: 3

Comments to the Author(s)

I think that it would have been necessary to examine sub samples of *L. santolla* by histology, including those do not display clinical signs of infection. Clinical signs as whitish nodules arising

from the sub-cuticular tissues are macroscopic signs of probably late-stage/patent infections (e.g. Field et al. 1992, Shields & Behringer 2004, Stentiford et al. 2010, 2014), and use of gross visual assessment methods may significantly underestimate the actual prevalence of *A. rohanei*. May be to save this point (since the sampling was already done), you could little explain in discussion section.

Prevalence associated with geographic locations was examined. Did you study it with respect to depth of samples?

No association between prevalence and 36 geographic location was detected, may be it could be explained by the different sampling sizes at each location (e.g. in Grupo Solari only 7 king crabs were collected whilst in Isla Toro 34 king crabs were collected). It might be discussed in the corresponding section. A table of these results presenting sampling size, sex proportion and their prevalence (besides the map) is suggested to be provided.

A co-habitation experiment with infected crabs and uninfected ones to explore potential for crab-crab transmission could be addressed.

Page7- Line111: Replace "oocytes" by "eggs"

Line 112: Idem

Page 10- Line 165: Replace "larger" by "higher"

What is the commercial size of *L. santolla*? It is suggested to provide this information.

Author's Response to Decision Letter for (RSOS-190682.R0)

See Appendix A.

RSOS-190682.R1 (Revision)

Review form: Reviewer 2 (Vladimir G. Dvoretzky)

Is the manuscript scientifically sound in its present form?

Yes

Are the interpretations and conclusions justified by the results?

Yes

Is the language acceptable?

Yes

Do you have any ethical concerns with this paper?

No

Have you any concerns about statistical analyses in this paper?

No

Recommendation?

Accept as is

Comments to the Author(s)

The authors have revised their ms adequately. I recommend accepting the ms for publication in Royal Society Open Science.

Review form: Reviewer 3

Is the manuscript scientifically sound in its present form?

Yes

Are the interpretations and conclusions justified by the results?

Yes

Is the language acceptable?

Yes

Do you have any ethical concerns with this paper?

No

Have you any concerns about statistical analyses in this paper?

No

Recommendation?

Accept as is

Comments to the Author(s)

The authors have clearly given careful and thorough attention to the comments from the reviews.

Decision letter (RSOS-190682.R1)

08-Sep-2019

Dear Dr Rodríguez,

I am pleased to inform you that your manuscript entitled "Widespread infection of *Areospora rohanae* in southern king crab (*Lithodes santolla*) populations across south Chilean Patagonia" is now accepted for publication in Royal Society Open Science.

on behalf of Prof Kevin Padian (Subject Editor)
openscience@royalsociety.org

Associate Editor Comments to Author :
Thank you for effectively responding to the referees' comments. We hope you'll consider submitting to RSOS again in future.

Reviewer comments to Author:
Reviewer: 2

Comments to the Author(s)
The authors have revised their ms adequately. I recommend accepting the ms for publication in Royal Society Open Science.

Reviewer: 3

Comments to the Author(s)
The authors have clearly given careful and thorough attention to the comments from the reviews.

Appendix A

Valdivia, August 09, 2019

Dra. Sara M. Rodríguez
Universidad Austral de Chile
Instituto de Ciencias Marinas y Limnológicas

Dr. Andrew Dunn
Journal Editor
The Royal Society

Dear Dr. Dunn,

Please find attached the corrected manuscript “Widespread infection of *Areospora rohanae* in southern king crab (*Lithodes santolla*) populations across south Chilean Patagonia” by SM Rodríguez, JC Uribe, SA Estay, M Palacios, R Pinochet, S Oyarzún and N Valdivia”.
In addition, we attached the Chilean government fishing permit used for catching *Lithodes santolla*.

Reviewers' Comments to Author:

Reviewer: 1

“Major:

This is a simple study that would be better presented as a shorter paper/note in a specialist journal. 57 cited references are overkill for a study that only assessed prevalence over a single sample of crabs from one time point (winter 2011). So little is known of this parasite (described in 2014), particularly its mechanisms of transmission, dispersal, seasonality etc., so most text hypothesizing mechanisms driving prevalence/distribution are supposition at best. I suggest the authors restructure the paper as a short report-the prevalence/distribution data is valuable, but the context needs to be changed.”

R: Thank you for this comment. We agree with Reviewer #1 on the fact that our study is based upon a single sample and one point of time. However, we would like to stress that this contribution is the first report of the spatial distribution of the infection, covering circa 800 km of the southeastern pacific shore. Moreover, the sample encompassed 3000 host individuals. Also, and perhaps more importantly, *Lithodes santolla* is an economically important natural resource of local communities in Magallanes and elsewhere. Thus, we see this contribution and a relevant benchmark to further develop a research agenda on the parasite-host dynamics of *Aerospora rohanae* and *L. santolla*. For those reasons, we have shortened the manuscript, reduced too speculative statements and sloppy citations, and also removed redundant literature (43 references), but with an eye in keeping the paper rather than the short note format. With this, we hope Reviewer #1 understand our concerns related to moving our paper to a note.

“It is not clear to this reviewer that the pathognomonic sign for infection of A. rohanae has been well established and tested, yet the gross presentation has been used as a diagnostic tool in the current study (lines 113-118). I have read Stentiford et al. (2014) study and nowhere in that paper are the number of crabs that were used to describe the parasite presented. PCR data was only generated from two independent samples. Stentiford et al do not indicate that no other infections can be associated to these symptoms in king crabs (incorrectly stated by the authors in lines 117-118). Further, cottage cheese disease (pathogen not described to species level) has previously been reported from king crabs in the northern hemisphere. Clearly further work needs to be carried out to establish, beyond doubt, that the presented symptoms are unique to A. rohanae infection.”

R: This is a welcome comment and we thank Reviewer #1 for pointing out this issue. In the revised manuscript, we have reviewed the literature to avoid sloppy citation and also to expand the discussion on the drawbacks of our work. Also, we have included an additional reference to discuss this issue (Shields and Behringer, 2004). Unfortunately, there is only one previous work on this parasite-host system, so we used the available information. Despite these issues, our manuscript still provides valuable information of the occurrence of the symptoms across a relatively large section of the SE Pacific. Please see lines 171 to 177 in the discussion and all changes in the marked copy.

Minor:

Line 35-36, expand on the stated significant association between prevalence and geographical location as there is no mention of this in the results section, page 8??

R: Done. We have expanded the result section to make clear that the geographic location had a very low predictive value for the infection.

Line 41-42, delete as speculation.

R: Done. The line was deleted.

Lines 67-68, FYI conversely, many crustaceans show a higher prevalence of pathogens in earlier life stages.

R: Thank you for pointing out this information. In order to simplify the theoretical background, we have removed this sentence from the introduction.

Line 71, rewrite to read “In the case”

R: Done. Thank you.

Line 81-82, better describe “depolymerization of the contractile apparatus” as I don’t understand what the authors mean. How do spores do this?

R: Thank you for pointing out this problem. We have removed the unnecessary sentence in the revised manuscript.

Line 86, do Lithodid crabs harbor diverse microsporidian parasites? I have only seen very few microsporidian species mentioned in the associated king crab literature? Can the authors elaborate?

R: Done. We clarify that there are not many microsporidian species that parasitized lithodid crabs. Yet, this is briefly mentioned in order to keep the manuscript as short as possible. Please see lines 75 to 77.

Line 105, what is a conical plot? Do the authors not mean "pots"? Provide mesh width of traps/pots.

R: Yes, thank you. We have standardized the name of the observation unit throughout the manuscript.

Line 172-173, there is NO data to suggest that A. rohanae transmission is by horizontal mechanisms. Stentiford et al. do not mention this anywhere in their paper. Alternate hosts may well be involved but this has not been studied.

R: Thank you for pointing out this example of sloppy citation. As part of the process of shortening the manuscript, the discussion of horizontal transmission was removed from the manuscript as it was too speculative.

Line 180, a single infected host could release millions of infective spores, not hundreds....

R: This sentence was properly adjusted in the revised manuscript. Please see line 167 to 168.

Line 189-190, no where in this study was the presence of A. rohanae established other than by gross presentation. It is completely unethical/unscientific to state that "L. santolla was infected only with A. rohanae" especially as the material the authors worked with was frozen at -80 deg (destroying tissues).

R: Thank you for pointing out this mistake. The misleading sentence was removed from the revised manuscript and now we explicitly acknowledge the limitation of our study. Please refer to our reply to the second major comment of Reviewer #1.

Reviewer: 2

Comments to the Author(s)

The authors of the manuscript "Widespread infection of *Areospora rohanae* in southern king crab (*Lithodes santolla*) populations across south Chilean Patagonia" have written a fairly clear and concise paper on a relevant topic. However, it needs some clarification. Here, I have recommended a number of minor revisions that

should be fixed in a revised version.

L. 33, L. 105. Change "conical plots" to "conical pots".

R: Done.

L. 34, L. 106. Change "an ample range" to "a wide range"

R: Done.

L. 59. Change "lowers" to "decreases"

R: Done.

L. 75 Change "little" to "low"

R: Done.

L. 87 Change "2000 and 4000 tons year⁻¹" to "2,000 and 4,000 t per year"

R: Done.

L. 106 Be consistent: change "traps" to "pots" or "pots" to "traps" throughout the text.

R: Done. In the revised manuscript, we use the term "pot" throughout the text.

L. 106 Change "50 and 200 meters" to "50 and 200 m"

R: Done.

L. 112 Delete ";

R: Done.

L. 119 Change "ranges" to "classes"

R: Done.

L. 160 Change "societal" to "socio-economic"

R: Done.

L. 164 Delete "and"

R: Done.

*L. 185 Change "and anomuran crab (*Eupagurus bernhardus*)" to "and the hermit crab *Eupagurus bernhardus*"*

R: Thank you. Please note that we have removed this sentence to shorten the manuscript.

L. 189 Change "pathogenesis they" to "their pathogenesis"

R: Done.

L. 196 Change "may shed some light" to "may provide data"

R: Done.

L. 211-212 Change "region *L. santolla* is the most important fishery" to "region, *L. santolla* support the most important fishery".

R: Done.

L. 217 Change "living king crabs" to "live king crabs"

R: Done.

L. 219 Change "socioeconomical" to "socio-economic"

R: Done.

L. 345 Change "Fish B-NOAA" to "Fish. Bull."

R: Done.

L. 413 Delete "in."

R: Done.

Page 24. In my opinion, for better presentation, the authors should delete the number of individuals for each size range and sex from their Figure 1. Optionally, this information can be provided in a separate table (in Materials and Methods).

R: Done, thank you. Fig. 4 was modified according to Reviewer #2' suggestion and an additional table (table 1) was incorporated in the revised manuscript.

Reviewer: 3

Comments to the Author(s)

*I think that it would have been necessary to examine sub samples of *L. santolla* by histology, including those do not display clinical signs of infection. Clinical signs as whitish nodules arising from the sub-cuticular tissues are macroscopic signs of probably late-stage/patent infections (e.g. Field et al. 1992, Shields & Behringer 2004, Stentiford et al. 2010, 2014), and use of gross visual assessment methods may significantly underestimate the actual prevalence of *A. rohanae*. May be to save this point (since the sampling was already done), you could little explain in discussion section.*

R: This is an interesting point, thank you. We have developed the discussion of the drawback of our study in lines 182-188 of the revised manuscript (see also our marked copy), including also potential confounding effects of other-than-Aerospora microsporidian on the observed prevalence. Nevertheless, we still believe that our study would be a valuable contribution to basic, life-history knowledge of microsporidian parasites and king crabs, as this is the first analysis of the spatial distribution of the parasite in a relatively large section of the SE Pacific.

Prevalence associated with geographic locations was examined. Did you study it with respect to depth of samples?

R: Thank you for this question. Unfortunately, and due to logistic onboard constraints, we were unable to accurately record sample depth. Yet, we used a

depth range that included most of king crab' sizes (i.e. 50 to 200 m) in order to assess size-dependent parasitosis.

No association between prevalence and 36 geographic location was detected, may be it could be explained by the different sampling sizes at each location (e.g. in Grupo Solari only 7 king crabs were collected whilst in Isla Toro 34 king crabs were collected). It might be discussed in the corresponding section. A table of these results presenting sampling size, sex proportion and their prevalence (besides the map) is suggested to be provided.

R: Thank you for pointing out this issue. In the revised manuscript, we have included the number of crabs of each sampling station as a predictive factor in the statistical model. The result remained qualitatively similar to the previous fit ($r^2 = 0.018$), showing a very poor explanatory power of crab and sample attributes. This information is now on lines 142-144. Also, a new table now includes the number of individuals (separated by sex) and prevalence. Please see also our marked manuscript.

A co-habitation experiment with infected crabs and uninfected ones to explore potential for crab-crab transmission could be addressed.

R: This is an interesting suggestion—many thanks. Indeed, we propose that further manipulative research is needed to better understand the dynamics in this parasite-host system. Please also refer to our first response to Reviewer #3's comments.

Page7- Line111: Replace "oocytes" by "eggs"

R: Done.

Line 112: Idem

R: Done.

Page 10- Line 165: Replace "larger" by "higher"

R: Done.

What is the commercial size of L. santolla? It is suggested to provide this information.

R: Done. The information of commercial size and sex was added on line 85. Thank!